# Automatic Analysis of Substantiation in Scientific Peer Reviews

**Yanzhu Guo**[1], **Guokan Shang**[4], **Virgile Rennard**[1,4], **Michalis Vazirgiannis**[1], **Chloé Clavel**[2,3]

[1]LIX, École Polytechnique, Institut Polytechnique de Paris, France
[2]LTCI, Télécom-Paris, Institut Polytechnique de Paris, France
[3]Inria, Paris, France [4]Linagora, France
`{yanzhu.guo, guokan.shang}@polytechnique.edu`
`virgile@rennard.org, mvazirg@lix.polytechnique.fr`
`chloe.clavel@telecom-paris.fr`

## Abstract

With the increasing amount of problematic peer reviews in top AI conferences, the community is urgently in need of automatic quality control measures. In this paper, we restrict our attention to substantiation — one popular quality aspect indicating whether the claims in a review are sufficiently supported by evidence — and provide a solution automatizing this evaluation process. To achieve this goal, we first formulate the problem as claim-evidence pair extraction in scientific peer reviews, and collect SubstanReview, the first annotated dataset for this task. SubstanReview consists of 550 reviews from NLP conferences annotated by domain experts. On the basis of this dataset, we train an argument mining system to automatically analyze the level of substantiation in peer reviews. We also perform data analysis on the SubstanReview dataset to obtain meaningful insights on peer reviewing quality in NLP conferences over recent years. The dataset is available at `https://github.com/YanzhuGuo/SubstanReview`.

## 1 Introduction

Peer review is an essential practice for gauging the quality and suitability of scientific papers in the academic publication process (Price and Flach, 2017). However, in recent years, this step has been widely criticised for its unreliability, especially in top Artificial Intelligence (AI) conferences (Tran et al., 2020). This is partially due to the surge in numbers of submitted papers and the shortage of domain experts fulfilling the requirements to serve as reviewers (Publons, 2018; Russo, 2021). This leads to increasing workload per reviewer and paper-vetting by less-experienced researchers (Ghosal et al., 2022), which consequently increases the likelihood of poor-quality reviews. Therefore, developments that can make the quality control process associated with peer reviewing more efficient, are likely to be welcomed by the research community (Checco et al., 2021).

To this end, our research aims to develop an automatic system to analyze the quality of peer reviews. Such a system would be of great value not only to chairs, who could use them to eliminate identified poor-quality reviews from the decision-making process, but also to reviewers, who might be instructed to improve the writing at the time of reviewing. They could be exploited by conference managers as well to analyze overall review quality and reviewer performance, in order to better organize the next review round.

Clearly, and according to the review guidelines of several top AI conferences, including ICLR, ICML, NeurIPS, ACL and EMNLP, reviews should be assessed from multiple perspectives (e.g., domain knowledgeability, factuality, clarity, comprehensiveness, and kindness). In this paper, we decide to focus on one specific quality dimension – **substantiation**, which has appeared in nearly all review guidelines, sometimes under different names such as *specificity*, *objectiveness*, or *justification*. This criterion states that a good review should be based on objective facts and reasoning rather than sentiment and ideology. Specifically, each subjective statement (**claim**) in the review should be backed up by details or justification (**evidence**). More discussion on substantiation will be provided in Section 3.1.

To progress towards the goal of automatically evaluating the level of substantiation for any given review, we employ an **argument mining** approach. Scientific peer reviewing can be viewed as an argumentation process (Fromm et al., 2021), in which reviewers convince the program committee of a conference to either accept or reject a paper by providing arguments. In our annotation scheme for arguments (see Section 4.2), the two basic components are claims and evidence. A substantiated argument is one where the claims are supported by evidence. Therefore, we formulate the task of **claim-evidence pair extraction** for scientific peer

reviews.

We release SubstanReview, a dataset of 550 peer reviews with paired claim and evidence spans annotated by domain experts. On the basis of this dataset, we develop an argument mining system to perform the task automatically, achieving satisfactory performance. We also propose SubstanScore, a metric based on the percentage of supported claims to quantify the level of substantiation of each review. Finally, we use the SubstanReview dataset to analyze the substantiation patterns in recent conferences. Results show a concerning decrease in the level of substantiation of peer reviews in major NLP conferences over the last few years. Our contributions are threefold:

1. We define the new task of claim-evidence pair extraction for scientific peer reviews and create the first annotated dataset for this task.

2. We develop a pipeline for performing claim-evidence pair extraction automatically while leveraging state-of-the-art methods from other well established NLP tasks.

3. We provide meaningful insights into the current level of substantiation in scientific peer reviews.

## 2   Related Work

**Review Quality Analysis.** The exponential growth of paper submissions to top AI conferences poses a non-negligible challenge to the peer reviewing process, drawing considerable research attention during recent years (Severin et al., 2022). Multiple peer review datasets have been released along with the corresponding papers, mostly taken from the OpenReview platform (Kang et al., 2018; Cheng et al., 2020; Fromm et al., 2021; Kennard et al., 2022). More recent efforts have been made to collect opted-in reviews that are not publicly accessible, often through coordination with the program committee (Dycke et al., 2023, 2022).

These resources have been used to carry out studies on peer review quality. Previous works on automatically evaluating scientific peer reviews have targeted the aspects of harshness (Verma et al., 2022), thoroughness (Severin et al., 2022), helpfulness (Severin et al., 2022) and comprehensiveness (Yuan et al., 2022). Yet, their methodologies are usually based on regression models trained with human annotated scores, which often lack in both generalizability and interpretability.

A quality aspect highly relevant to substantiation, is "justification", previously studied by Yuan et al. (2022) as an evaluation measure for their automatic review generation model. More specifically, they state that *"a good review should provide specific reasons for its assessment, particularly whenever it states that the paper is lacking in some aspect"*. However, their evaluation protocol for justification relies solely on human annotators. Our work is the first one to automatically assess the substantiation level of scientific peer reviews. More importantly, we do not only provide a final quantitative score, but also highly interpretable claim-evidence pairs extracted through an argument mining approach.

**Argument Mining.** Lawrence and Reed (2019) define the task of argument mining as the automatic identification and extraction of argument components and structures. The state-of-the-art in argument mining was initially based on feature engineering, while more recent methods rely on neural networks (Ein-Dor et al., 2020), especially following the introduction of the transformer architecture. Previous works applying argument mining to peer reviews typically focus on identifying argumentative content and classifying it. Hua et al. (2019) introduced the AMPERE dataset containing 400 reviews annotated for proposition segmentation and proposition classification (evaluation, request, fact, reference, quote, or non argument) and trained neural models to perform the two tasks. Similarly, Fromm et al. (2021) performed annotations on 70 reviews for supporting arguments, attacking arguments and non arguments, and trained a BERT model for the tasks of argumentation detection and stance detection. None of these efforts take into account the structure of arguments, making our work the first to examine the link between different argument components in scientific peer reviews.

## 3   Task Formulation

In this section, we first take a closer look at substantiation and its definition. We then formulate its estimation as the claim-evidence pair extraction task. Finally, we introduce fundamental concepts in argumentation theory and explain how the claim-evidence pair extraction task can be tackled by an argument mining approach.

### 3.1   Defining Substantiation

While substantiation is only one of the criteria for a good review, it is a fundamentally important one.

The paper proposes a method to train models for Chinese word segmentation (CWS) on datasets having multiple segmentation criteria.

-Strengths:

1. [Multi-criteria learning is interesting and promising.]$_{claim\_pos\_1}$

2. [The proposed model is also interesting and achieves a large improvement from baselines.]$_{claim\_pos\_2}$

-Weaknesses:

1. [The proposed method is not sufficiently compared with other CWS models.]$_{claim\_neg\_1}$ [The baseline model (Bi-LSTM) is proposed in [1] and [2]. However, these model is proposed not for CWS but for POS tagging and NE tagging. The description "In this paper, we employ the state-of-the-art architecture ..." (in Section 2) is misleading.]$_{evidence\_neg\_1}$

2. [The purpose of experiments in Section 6.4 is unclear.]$_{claim\_neg\_2}$ [In Sec. 6.4, the purpose is that investigating "datasets in traditional Chinese and simplified Chinese could help each other." However, in the experimental setting, the model is separately trained on simplified Chinese and traditional Chinese, and the shared parameters are fixed after training on simplified Chinese.]$_{evidence\_neg\_2}$ What is expected to fixed shared parameters?

- General Discussion:

The paper should be more interesting if there are more detailed discussion about the datasets that adversarial multi-criteria learning does not boost the performance.

[1] Zhiheng Huang, Wei Xu, and Kai Yu. 2015. Bidirectional lstm-crf models for sequence tagging. arXiv preprint arXiv:1508.01991.

[2] Xuezhe Ma and Eduard Hovy. 2016. End-to-end sequence labeling via bi-directional lstm-cnns-crf. arXiv preprint arXiv:1603.01354.

Table 1: Example of an annotated peer review from the SubstanReview dataset. We select this particular review as an example due to its straightforward organization, where supporting evidence directly follows the claims. However, it should be noted that many other reviews have much more complex structures, rendering the task of claim-evidence pair extraction challenging.

Most of the times, paper authors are not unwilling to accept negative opinions about their work. However, if the argument is only based on subjective sentiments and no further supporting evidence, it is unlikely that the argument provides a fair evaluation that can lead to an appropriate acceptance/rejection decision. Moreover, the purpose of peer reviewing is not only to make the final decision but also to provide constructive feedback for the authors to eventually improve their work. This purpose cannot be achieved without sufficient evidence substantiating each point made in the review.

Compared to other criteria such as factuality, comprehensiveness, or domain knowledgeability, the analysis of substantiation is more straightforward. It does not necessitate a deep understanding of the paper under review. In fact, in our analysis of substantiation, our sole concern is *whether each subjective statement has supporting evidence but not whether the supporting pieces of evidence are factually correct*. Evaluating the correctness of evidence is left for future work on the dimension of factuality for peer reviews. However, the annotations still need to be carried out by domain experts who have a general understanding of AI research and the context of scientific peer reviews.

In short, we define substantiation of a scientific peer review as the percentage of subjective statements that are supported by evidence. Therefore, we propose and formulate the task of claim-evidence pair extraction for scientific peer reviews.

## 3.2 Claim-Evidence Pair Extraction

The task of claim-evidence pair extraction is separated into two steps: claim tagging and evidence linkage. Previous works on proposition segmentation have shown that segmenting peer reviews by sentence boundaries or discourse connectives do not yield optimal results (Hua et al., 2019). Therefore, we do not specify any predefined boundaries for claim or evidence spans. Both steps are performed at the token level. An example with annotated claim-evidence pairs is shown in Table 1.

**Claim Tagging.** The goal of this step is to identify all the subjective statements in a given review. Such statements include evaluation of the novelty, the soundness or the writing of the paper, etc. The definition of a subjective statement will be further elaborated in Section 4.2. The subjective statements are further divided by their polarity. Claims supporting the acceptance of a paper are considered positive while those attacking it are considered negative. Therefore, the subtask of claim tagging is formulated as sequence labeling with positive

and negative types. We adapt the BIO (Beginning, Inside, Outside) encoding scheme, resulting in 5 possible classes for each token (B-claim_positive, I-claim_positive, B-claim_negative, I-claim_positive and O).

**Evidence Linkage.** The evidence linkage step follows the claim tagging step. The goal of this step is to select a contiguous span of text from the review as supporting evidence for each retrieved claim, if such evidence exists. Formally, for each retrieved claim $C = (c_1, c_2, \ldots, c_{|C|})$, we concatenate it with the full review $R = (r_1, r_2, \ldots, r_{|R|})$ into a single sequence $S = ([CLS]c_1 c_2 \ldots c_{|C|}[SEP]r_1 r_2 \ldots r_{|R|})$. The task is thus to predict the evidence span boundary (start and end token position). We observe the similarity between this task and extractive question answering (QA). In both cases, the goal is to extract the most relevant text span (answer/evidence), given the context (article/review) and key sentences (question/claim). We therefore follow the QA model architecture proposed by Devlin et al. (2019), pass the concatenated sequence to a pre-trained transformer encoder and train two linear classifiers on top of it for predicting the start and end token position. For claims without supporting evidence, we simply set the answer span to be the special token [CLS]. We make the choice of first tagging claims and then linking a piece of evidence to each claim instead of extracting claims/evidence separately and then determining their relations. This way we ensure that each piece of evidence is dependent on a claim, since evidence cannot exist alone by definition.

### 3.3 Claim-Evidence Pair Extraction and Argumentation Theory

Argumentation aims to justify opinions by presenting reasons for claims (Lawrence and Reed, 2019). Scientific peer reviewing can be understood as a process of argumentation where reviewers need to justify their acceptance/rejection recommendations for a paper. We ground the task of claim-evidence pair extraction within the framework of argumentation theory following Freeman's model of argument (Freeman, 2011b,a). Freeman's model integrates Toulmin's model (Toulmin, 2003) and the standard approach (Thomas, 1973). Different from Toulmin's model, it proposes to analyze arguments *as product* rather than *as process*. As a result, it is more applicable to real-life arguments

and commonly exploited in computational linguistics settings (Lopes Cardoso et al., 2023).

The main elements of Freeman's model are *conclusions* and *premises*. The *conclusion* is a subjective statement that expresses a stance on a certain matter while *premises* are justifications of the *conclusion*. In the context of claim-evidence pair extraction, we define **claim** to be the *conclusion* and **evidence** as *premise*. Freeman (2011a) also proposes *modality* as an argument component indicating the strength of the argumentative reasoning step. *Modality* is often integrated into the *conclusion* or not present at all in practical arguments, therefore we do not model it individually but integrate it in the claim span. The last type of argument component is *rebutting/undercutting defeaters*. They are irrelevant to our analysis of substantiation and are thus not taken into consideration.

## 4 SubstanReview Dataset

In this section, we introduce SubstanReview, the first human annotated dataset for the task of claim-evidence pair extraction in scientific peer reviews. We first discuss the source of the reviews and then elaborate on the annotation process. Finally, we provide a statistic overview of our annotated dataset.

### 4.1 Data Source

The raw reviews used for annotation are taken from the NLPeer dataset (Dycke et al., 2023). NLPeer is the most comprehensive ethically sourced corpus of peer reviews to date. For all included reviews, both the corresponding author and reviewer have agreed to opt in. We use a subpart of NLPeer with reviews issued from NLP conferences (*CoNLL 2016*, *ACL 2017*, *COLING 2022* and *ARR 2022*[1]). We deliberately choose to only include NLP reviews because conferences in other sub-domains of AI generally vary a lot in review style, which might negatively impact the final system's performance, given the limited amount of total annotation capacity. We randomly select 50% of all available reviews for each of the different conferences, resulting in an annotated dataset of 550 reviews.

We do not make use of reviews collected without explicit consent through the OpenReview platform, which leads us to disregard datasets from previous

---

[1]ARR stands for ACL Rolling Review, all reviews included in *ARR 2022* are from papers later accepted at *ACL 2022* or *NACCL 2022*.

works (Hua et al., 2019; Fromm et al., 2021). These datasets are not clearly licensed, which might pose ethical and legal issues in the long term.

## 4.2 Annotation Study

In this section, we define our annotation scheme, introduce the annotation process and examine the disagreement between different annotators.

**Annotation Scheme.** Our annotation scheme is based on the argumentation model discussed in Section 3.3. The resulting scheme contains the following two argumentative components:

(1) Claim: Subjective statements that reflect the reviewer's evaluation of the paper. For defining subjectivity, we follow the notion of Wiebe et al. (2005): Subjectivity can be expressed either explicitly by descriptions of the writer's mental state or implicitly through opinionated comments. They are further separated by polarity into positive and negative classes.

(2) Evidence: Justifications of the subjective statements that serve as claims. For example, in the context of a reviewer pointing out a weakness of the paper, the premise could be specific examples of the problem, reasoning on why this is problematic or suggestions for solving the problem.

The complete annotation guidelines can be found in Appendix A.

**Inter-annotator Agreement.** We use Krippendorf's unitizing alpha[2] (Krippendorff et al., 2016) to calculate inter-annotator agreement (IAA). The $_u\alpha$-coefficient quantifies the reliability of partitioning a continuum into different types of units. In our case, there are 5 types of units in total (positive claims, negative claims, positive evidence, negative evidence and non). The task grows more difficult as the number of types increases.

**Annotation Rounds.** All annotations were done with the open-source data labeling tool doccano[3]. We held two rounds in total.

Pilot Round. The pilot study was carried out with fourteen annotators following an initial version of annotation guidelines. It was conducted on the PeerRead dataset (Kang et al., 2018) since the more comprehensive NLPeer dataset was not yet released back then. This round of annotations only led to a moderate $_u\alpha = 0.367$. The annotations from

[2] https://mathet.users.greyc.fr/agreement/
[3] https://doccano.github.io/doccano/

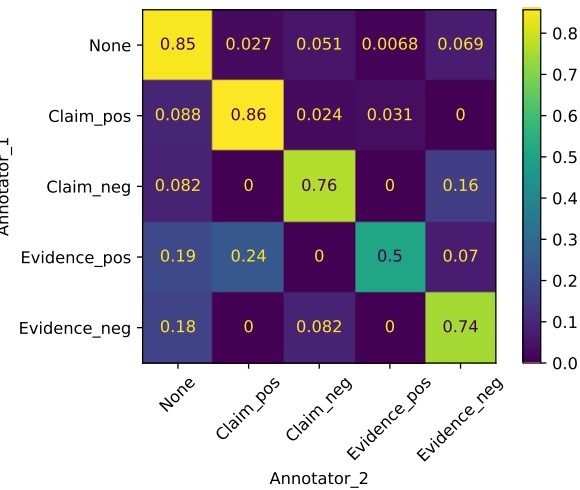

Figure 1: Confusion matrix (normalized by row) between annotations by annotator_1 and annotator_2 on reviews labeled by all three annotators (10% of Substan-Review).

this round were only used to refine the annotation guidelines and not included in the final Substan-Review dataset. More results from this annotation round can be found in Appendix B. We believe that the unsatisfactory IAA is due to the high number of annotators which naturally amplifies the bias (Lopes Cardoso et al., 2023). Moreover, the annotators only went through a 2 hour training session, thus not qualifying as experts for this task. We conclude that for such complicated annotation tasks involving argumentation theory, it is better to employ a small number of expert annotators rather than a high number of non experts.

Main Round. Following the pilot round, our main annotation study was carried out by three expert annotators who are coauthors of this paper. They are all graduate/post-graduate NLP researchers proficient in academic English, familiar with the context of scientific peer reviews and argumentation theory. They all participated in creating the annotation guidelines and supervising the pilot annotation round. They later had several meetings to identify factors causing disagreement during the pilot round and further refined the guidelines.

10% of the dataset (55 reviews) was used to calculate inter-annotator agreement (IAA). These 55 reviews were labeled independently by each of the three annotators, resulting in an Krippendorf's unitizing alpha (Krippendorff et al., 2016) of $_u\alpha = 0.657$. While other works that simply partition texts into argument and non argument types might achieve even higher IAA, our task is

| | #Claims | | | %Supported claims | | | len(Review) | #Reviews |
|---|---|---|---|---|---|---|---|---|
| | Pos | Neg | All | Pos | Neg | All | - | - |
| CoNLL 2016 | 2.01 | 1.94 | 2.95 | 27.97 | 87.03 | 51.82 | 483 | 19 |
| ACL 2017 | 2.62 | 2.91 | 5.54 | 26.66 | 78.58 | 47.72 | 499 | 134 |
| COLING 2020 | 2.70 | 2.78 | 5.38 | 35.04 | 74.71 | 45.43 | 512 | 56 |
| ARR 2022 | 2.73 | 2.25 | 4.98 | 30.37 | 75.54 | 44.69 | 472 | 341 |

Table 2: Statistics of the SubstanReview dataset, reported values are the mean over all reviews. **#Claims** stands for the *number of claims*, **%Supported claims** stands for the *percentage of claims that are paired with evidence*.

at a higher difficulty level. Compared to similar efforts annotating refined argument structures (Rocha et al., 2022) ($_u\alpha = 0.33$), our IAA score is significantly improved.

The rest 90% of the dataset (495 reviews) was randomly split into three equal portions and each annotated by only one annotator.

**Inter-annotator Disagreement.** The token level confusion matrix between annotations (annotator_1, annotator_2) is shown in Figure 1. The confusion matrices between (annotator_1, annotator_3) and (annotator_2, annotator_3) are highly similar and therefore omitted here. We see that the main disagreement arises between claims and evidence of the same polarity.

### 4.3 Statistics and Insights

We present several statistics of our annotated SubstanReview dataset in Table 2. For all conferences, there exists the same trend that more positive subjective claims are detected compared to negative ones. In contrast, the percentage of supported negative claims is higher than the percentage of supported positive claims. This is in line with current review practices since most reviewers believe it to be more relevant to provide specific reasoning when stating that a paper is lacking in some aspect (Yuan et al., 2022).

Conferences included in our analyzed datasets range from 2016 to 2022. We observe that the average length of reviews are generally on the same level for all NLP conferences, with COLING 2020 the longest and ARR 2022 the shortest. For the proportion of supported claims, there is a continuous decrease from CoNLL 2016 to ARR 2022. This observation can be understood to correspond to the surge of problematic peer reviews in recent years. Our finding is consistent with the one reported by Tran et al. (2020), that the peer review process has gotten worse over time for machine learning conferences.

### 4.4 SubstanScore

We propose a quantitative score measuring the overall substantiation level of a given peer review

$$SubstanScore = \\ \%supported\_claims \times len(review).$$

As defined in Section 3.1, a well substantiated review is one where a high proportion of subjective claims are supported by evidence. If a review does not contain any subjective claims, we consider it to be fully objective and assume %supported_claims=100%. However, a short review with few or no subjective claims may also contain limited substantial information overall, even if %supported_claims is high. To address this bias, we multiply %supported_claims by the review length (number of words in the review).

During the annotation study, in addition to marking the spans, we also asked each annotator to rate the substantiation level of each review on a 3 point Likert scale, with 3 representing the strongest level of substantiation.

We calculate the correlation between SubstanScore and the human annotated substantiation scores. We obtain Spearman's $\rho = 0.7568$ ($p = 6.5 \times e^{-20}$), i.e., a positive correlation between SubstanScore and human judgements.

We also calculate correlations between SubstanScore and %supported_claims or len(review) separately. Both give worse correlation than the combined SubstanScore.

## 5 Experiments and Results

We tackle the claim-evidence pair extraction task formulated in Section 3.2 and construct a benchmark for the SubstanReview dataset. The claim tagging is treated as a token classification task while the evidence linkage is approached as a question-answering task. We solve both of these tasks by fine-tuning pretrained transformer encoders

|  | Claim Tagging | | | Evidence Linkage | |
|---|---|---|---|---|---|
|  | Precision | Recall | F1 | Exact match | F1 |
| BERT | 41.01 | 52.40 | 46.01 | 43.17 | 78.15 |
| RoBERTa | **52.00** | **59.77** | **55.61** | 48.90 | 80.24 |
| SciBERT | 39.66 | 54.48 | 45.91 | 46.69 | 80.05 |
| SpanBERT | 53.67 | 38.81 | 36.12 | **64.31** | **82.07** |
| Baseline | 15.78 | 9.890 | 12.16 | 3.456 | 10.78 |

Table 3: Results for the claim-evidence pair extraction task. For the best performing models, we use a two-sided t-test to confirm that the results are statistically significant ($p < 5\%$).

(Vaswani et al., 2017), with added task-specific classification layers. To deal with the limited input sequence length of models, we split the input reviews into chunks with a maximum length of 512 tokens (in case longer than the limitation). For evidence linkage, to ensure that the start and end tokens of a piece of evidence are in the same chunk, we also add a sliding window with a stride of 128.

## 5.1 Models

We test the original BERT model along with other alternatives optimized or adapted to the domain/nature of our task:

**BERT** (Devlin et al., 2019) is the most popular transformer-based model, its base version has 12 encoder layers with 110M parameters in total while its large version has 24 encoder layers with 340M parameters. We use the large version[4] in our experiments.

**RoBERTa** (Liu et al., 2019) is an optimized version of BERT trained with a larger dataset using a more effective training procedure. We also use the large version of RoBERTa[5].

**SciBERT** (Beltagy et al., 2019) uses the original BERT architecture and is pretrained on a random sample of 1.14M papers from Semantic Scholar, demonstrating strong performance on tasks from the computer science domain. It is released in the base version [6].

**SpanBERT** (Joshi et al., 2020) modifies the pretraining objectives of BERT to better represent and predict spans of text. It demonstrates strong performance on tasks relying on span-based reasoning such as extractive question answering. We use its

large version[7].

## 5.2 Experimental Setup

We make a 80/20 split of the dataset for training and testing. Both the train and test splits can be found in the supplementary material. Hyperparameters are tuned using 5-fold cross-validation on the training set. Training is done over 10 epochs with a batch size of 8 and early stopping. The AdamW optimizer with 0.01 weight decay is used. Each model is trained 5 times with different randomization and the mean results are reported. All experiments are conducted on two 48GB NVIDIA RTX A6000 GPUs. The average training time is around 7 minutes for the base version model SciBERT and around 11 minutes for the large version models (BERT, RoBERTa and SpanBERT).

## 5.3 Results

In addition to the transformer-based models finetuned on SubstanReview, we also compute a baseline for both of the subtasks. For claim tagging, we first segment the reviews into sentences and pass them to a sentiment classifier[8] (Barbieri et al., 2020). Tokens in sentences predicted with positive sentiment will be identified as positive claims, tokens in sentences predicted with negative sentiment will be assigned as negative claims, tokens predicted with neutral sentiment will not be assigned part of claims. For evidence linkage, we select the sentence with the highest BERTScore (Zhang* et al., 2020) similarity to each claim as its evidence. Results are shown in Table 3.

**Claim Tagging.** We report the macro-averaged Precision, Recall and F1 scores for the claim tag-

---

[4] https://huggingface.co/bert-large-uncased
[5] https://huggingface.co/roberta-large
[6] https://huggingface.co/allenai/scibert_scivocab_uncased

[7] https://huggingface.co/SpanBERT/spanbert-large-cased
[8] https://huggingface.co/cardiffnlp/twitter-roberta-base-sentiment

|            | Claim | | Evidence | |
|------------|-------|-------|-------|-------|
|            | Pos   | Neg   | Pos   | Neg   |
| Precision  | 78.89 | 81.34 | 24.78 | 75.02 |
| Recall     | 53.79 | 67.48 | 56.79 | 33.06 |
| F1         | 63.78 | 73.56 | 34.50 | 45.23 |

Table 4: Combined results of the two subtasks, taking error propagation into account.

ging subtask. These metrics are designed specifically to evaluate sequence tagging (Ramshaw and Marcus, 1995; Nakayama, 2018), they are very stringent as they only consider a predicted span as true positive if both the class (positive_claim/negative_claim) and segmentation exactly match the ground truth. The baseline is shown to give poor performance, demonstrating the difficulty of this task. Although SciBERT and SpanBERT bring considerable improvements in performance compared to the original BERT, they are not able to yield as significant of a gain as RoBERTa. We thus use the fine-tuned RoBERTa model for claim tagging in our final argument mining system.

**Evidence Linkage.** We provide the Exact Match (EM) and F1 scores for the evidence linkage subtask. These are the common metrics to report for tasks based on extractive QA. EM measures the percentage of samples where the predicted evidence span is identical to the correct evidence span, while F1 score is more forgiving and takes into account the word overlap ratio between predicted and ground truth spans. Our models still achieve a significant improvement over the baseline. Different from claim tagging, SpanBERT obtains the best results on this subtask. We choose to include the fine-tuned SpanBERT model for evidence linkage in our final system.

**Combined Performance.** We analyze the performance of the whole pipeline, combining the best performing models for each subtask. In Table 4, we show the token level classification results for each class. We observe that the combined pipeline performs better for claims than evidence, despite evidence linkage achieving better results than claim tagging when performed independently. This originates from error propagation, as evidence linkage is performed on top of predictions by the claim tagging model instead of the ground truth. We also find negative claims and evidence to be better

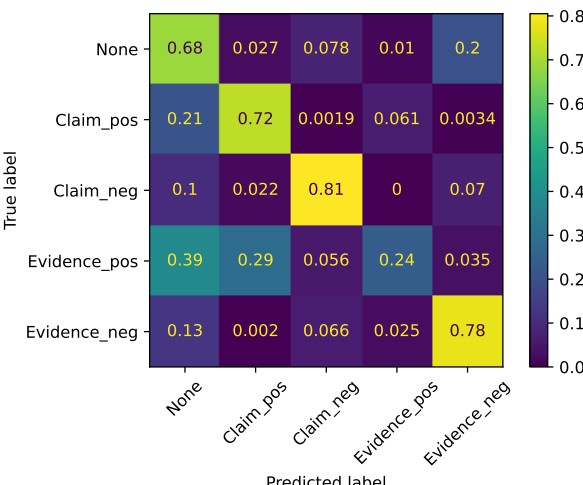

Figure 2: Confusion matrix (normalized by row) between model predictions and human annotations on the test set (20% of SubstanReview).

extracted than positive ones.

### 5.4 Error Analysis

The confusion matrix between model predictions and the ground truth is shown in Figure 2. It is similar to the confusion matrix of annotations shown in Figure 1. Both human annotators and the model appear to be struggling to make the same distinctions in the claim-evidence pair extraction task. Error propagation also remains an important challenge as evidence tokens are much more often misclassified compared to claim tokens. In future work, we plan to explore a single-step instruction tuning approach to mitigate this problem.

### 5.5 Comparison with Prompt-based Methods

Recently, the widespread recognition and usage of large language models (LLMs) has drawn the paradigm in NLP research to prompting methods (Liu et al., 2023). Therefore, we complete our work by providing a case study of using ChatGPT[9] (Ouyang et al., 2022) to tackle our claim-evidence pair extraction task.

The examples in Appendix C demonstrate that ChatGPT is not able to achieve satisfactory performance, with both specifically designed zero-shot and few-shot prompts, which highlights the need of our annotated data for instruction-tuning, and the superiority of classical task-specific fine-tuned models as proposed in our work.

In future work, our dataset could also be used for

[9] https://openai.com/blog/chatgpt/

instruction tuning and in-context learning, where only a small amount of high-quality human curated data is required (Zhou et al., 2023).

# 6 Conclusion

In this paper, we focus on automatically analyzing the level of substantiation in scientific peer reviews. We first formulate the task of claim-evidence pair extraction, comprised of two subtasks: claim tagging and evidence linkage. We then annotate and release the first dataset for this task: SubstanReview. We perform data analysis on SubstanReview and show interesting patterns in the substantiation level of peer reviews over recent years. We also define SubstanScore, which positively correlates with human judgements for substantiation. Finally, we develop an argument mining system to automatically perform claim-evidence pair extraction and obtain a great increase in performance over the baseline approach. We hope that our work inspires further research on the automatic analysis and evaluation of peer review quality, which is an increasingly important but understudied topic.

# Limitations

The main limitation of our work lies in the restricted scope of peer reviews included in our dataset.

Although attracting increasing amounts of attention recently, clearly licensed datasets of peer reviews are still very scarce and all of them are based on a donation-based workflow. This means that we only have access to peer reviews of which both the paper author and reviewer have expressed explicit consent to opt in. This introduces bias in our dataset, as reviewers are more likely to give consent to publish their reviews if they are confident in the quality of the review. Therefore, the quality of reviews (including the level of substantiation) included in our annotated dataset might be skewed towards the higher end. Systems trained on these data may encounter problems when applied in real-world scenario, where the review qualities are more balanced.

Given the high level of expertise required and significant amounts of efforts involved to annotate peer reviews, we could not perform the annotations on a larger scale. Thus, we have restricted our dataset to only include peer reviews from NLP conferences, leading to the potential lack of domain generalizability of our argument mining system.

Collecting peer review datasets with more representative distributions and more diverse domains should be considered for future work.

# Ethics Statement

All peer review data involved in this paper is used under explicit consent granted by their creators. All datasets are published under the CC0 or CC-BY license. We only use these datasets for research purposes which is consistent with their intended use. While peer review data is highly sensitive and may contain somehow offensive content, the employed datasets are anonymous and cannot be linked to individual people.

During the annotation procedure, we have notified the annotators of the intended use of the dataset and obtained their full consent to publish the annotations. All annotators are paid fair wage for their efforts.

We would like to state that the argument mining system resulting from our project is still a research prototype and should not be utilized for practical evaluations, especially when decision making is involved. Its validity and fairness still need to be extensively tested in real-world settings.

# Acknowledgements

We thank the ANR-TSIA HELAS chair for supporting the first and fourth authors.

We also thank the annotators who participated in the pilot round of annotation study: Hadi Abdine, Johannes Lutzeyer, Ashraf Ghiye, Christos Xypolopoulos, Ayman Qabel, Moussa Kamal Eddine, Iakovos Evdaimon, Sissy Kosma, Yassine Abbahaddou, Giannis Nikolentzos and Michalis Chatzianastasis.

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

# A    Annotation Guidelines

## Annotating Peer Reviews to Evaluate Substantiation

With the increasing amount of problematic peer reviews in top AI conferences, the community is urgently in need of automatic quality control measures. The goal of our project is to evaluate the substantiation of scientific peer reviews. We aim to develop an argument mining system that can automatically detect claims that are **vague, generic and unsubstantiated**. For this purpose, we ask all annotators to participate in the creation of the SubstanReview dataset, which will eventually be publicly released for research purposes. The task is to highlight **claims** in reviews from the NLPeer dataset, as well as the **evidence** of each claim (if they exist).

**[Freeman's model of argument]**

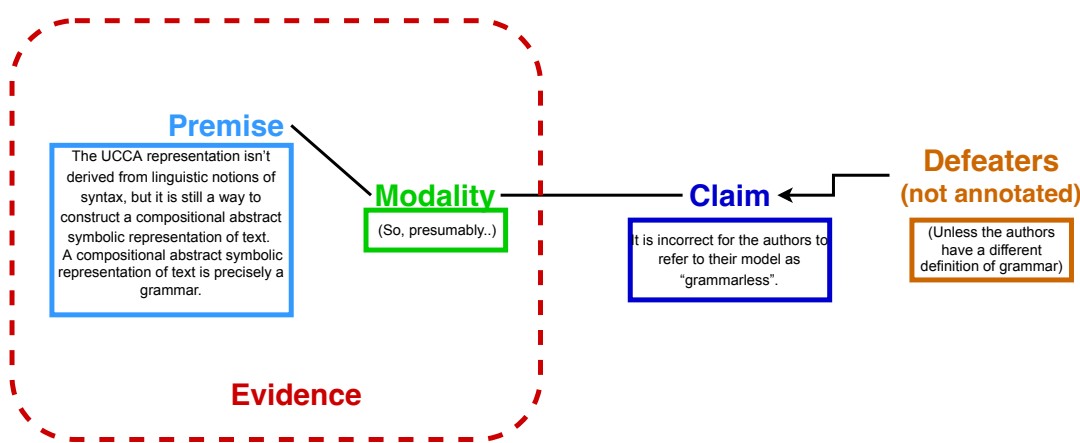

Figure 3: An example of the structure of an argument extracted from a review annotated according to Freeman's model of argument.

The main elements of Freeman's model are *conclusions* and *premises*. The *conclusion* is a subjective statement that expresses a stance on a certain matter while *premises* are justifications of the *conclusion*. In the context of claim-evidence pair extraction, we define **claim** to be the *conclusion* and **evidence** as *premise*. Freeman also proposes *modality* as an argument component indicating the strength of the argumentative reasoning step. *Modality* is often integrated into the *conclusion* or not present at all in practical arguments, therefore we do not model it individually but let it take part in the claim span. The last type of argument component is *rebutting/undercutting defeaters*. They are irrelevant to our analysis of substantiation and thus not taken into consideration.

**[Major Claims vs. Claims]**

We model the argumentation structure of a review as a two-level tree structure. The major claim (level 0) is the root node and represents the reviewer's general standpoint on whether the paper should be accepted. The major claim should not be annotated. We aim at annotating the more specific arguments which either support or attack the major claim. These arguments are further separated into claims (level 1) and evidence (level 2).

**[Task overview]**

**1**. Annotate **Claims** (*Def:* Subjective statements that convey the reviewer's evaluation of the research, related to the paper acceptance/rejection decision-making process).

- Claims are characterized by their subjectivity. Subjectivity can be expressed explicitly by descriptions of the writer's mental state, such as in " I'm not convinced of many of the technical details in the paper " and in " I disagree a little bit here ". It can also be expressed implicitly by opinionated descriptions of the work as in " There is no clear definition given for what this means "

and " This paper lacks quite a bit on comparison with existing work ". Both the adjective "clear" and the verb "lack" indicate subjective opinions and need to be further justified.

- Two evaluations should be separated if they are evaluating different aspects of the paper.

- Evaluations should be numbered according to their order of appearance.

**2**. Annotate **Evidence** (*Def:* Justifications of the subjective statements that serve as claims. For example, in the context of a reviewer pointing out a weakness of the paper, the premise could be specific examples of the problem, reasoning on why this is problematic or suggestions for solving the problem.).

- Not all claims have an evidence.

- Evidence do not have to be correct.

- Evidence can appear both before and after the Claim.

- Evidence should be numbered according to the claim they support.

**3**. In the comments section of each review

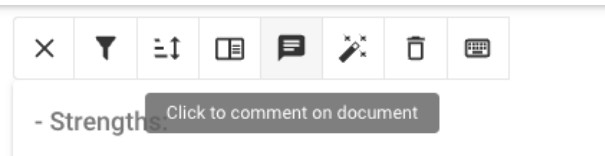

Figure 4: Comment button in the toolbar on the top of the user interface.

- Rate the **substantiation** of each review on a scale from 1 to 5.
  1 (Poor) – The vast majority of the claims in the review are vague, generic, and unsubstantiated. No comments that specifically relate to the content/substance of the work.
  2 (Insufficient) – Few of the claims in the review are substantiated, i.e., supported by evidence.
  3 (Average) – The review contains both valid and supported claims as well as some unsubstantiated statements and opinions.
  4 (Sufficient) – Most important claims are well justified although some claims and opinions require further substantiation.
  5 (Solid) – The vast majority of the claims are meaningful and well supported with evidence; reviewer's opinions are well argued for.

- Rate the **annotation difficulty** of each review on a scale from 1 to 3.
  1 (easy); 2 (medium); 3 (difficult)
  Write the substantiation score first and the annotation difficulty score second.
  The two scores should be separated by a **semicolon** (;)

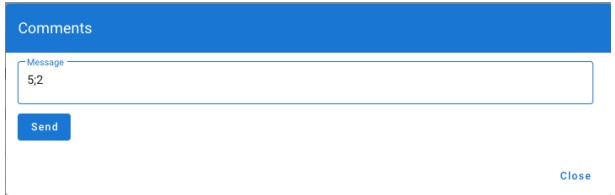

Figure 5: An example of a comment with two scores separated by a semicolon.

**[Further clarifications]**
**1**. Annotations are done at the **token level**.

- You can annotate arbitrary spans of text without taking into consideration any punctuation marks.

- Favor longer spans of text whenever possible, do not only annotate keywords.

**2**. Claims and evidence spans can coexist within the same sentence.
Example: "However, [ the major limitation the reviewer captures from the paper ](evaluation) [ is that the BCD is only used during the test stage ](justification)."

**3**. A lot of content in the text might remain unannotated. For example, facts that do not justify an explicit evaluation should not be annotated. Reactions to rebuttals should also not be annotated.

**4**. Don't forget to click on the leftmost button in the toolbar to mark that an annotation is completed.

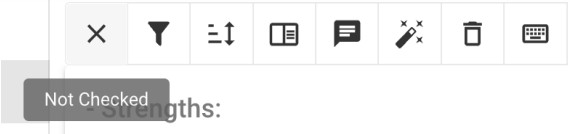

Figure 6: Check button in the toolbar on the top of the user interface.

## B  Pilot Annotation Round

We introduce our pilot round of annotation. Although it only led to moderate IAA ($_u\alpha = 0.367$), it helped us better understand the underlying difficulties and improve our annotation methodology accordingly.
**Annotation Process.** All annotators have gone through a 2 hour training session before proceeding with the annotations. Each annotator is also asked to complete three annotation samples individually and given feedback on their performance, verifying that they have understood correctly the annotation principles. Each review is randomly assigned to three annotators resulting in an average of 67 reviews per annotator. Annotators report an average of 4 hours to complete the assigned task. These hours count as normal working hours under their research contracts and are paid well above the local minimum wage. However, this round of annotations only led to a moderate annotator agreement (IAA) with Krippendorf's unitizing alpha $_u\alpha = 0.367$.
**Post-processing.** To aggregate annotations from different annotators into the final dataset, a post-processing step is required. We build a consensus between different annotators and obtain a unique annotated span for each claim and evidence.

For annotations of claims, we assign a label to a token if at least two of the three annotators have chosen the same label (majority voting).

The final evidence annotations are built upon the aggregated set of claim annotations. To solve the problem that the aggregated claims may not exactly match the claims annotated by each annotator, we examine the percentage of word overlap between them. For each aggregated claim $C_i$ in the final dataset, if a claim $C_j^a$ (annotated by annotator $a \in \{1, 2, 3\}$) has at least $60\%$ percent word overlap with $C_i$, then we consider it to correspond to $C_i$. The evidence $E_j^a$ linked to $C_j^a$ by annotator $a$ is thus also considered linked to the claim $C_i$. Just like with claim aggregation, all the evidence spans linked to $C_i$ are aggregated via majority voting.
**Dataset Statistics.** We present several statistics in Figure 7. The annotated dataset contains 314 reviews with an average length of 518 words. Comparing between claims and evidence, we observe that the average length of evidence (35) is significantly longer than that of claims (13). When comparing positive and negative classes, we find that the average number of negative claims per review (1.80) is slightly higher than positive ones (1.36) and that the average length of negative claims (14) is longer than that of positive claims (11). The average length of evidence for both negative and positive claims is almost the same (30 and 29 respectively). However, the range of length for negative evidence is much wider (up to 100). The percentage of supported negative claims is 84.91% while it is only 61.24% for positive ones. As a general trend, reviewers tend to provide more detailed explanations for their negative evaluations.

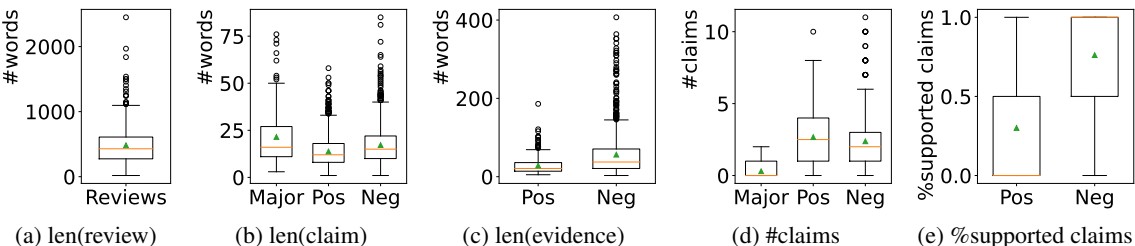

(a) len(review)    (b) len(claim)    (c) len(evidence)    (d) #claims    (e) %supported claims

Figure 7: Box plot of statistics for the SubstanReview dataset. Orange lines represent the median, green triangles represent the mean.

## C  Case study on ChatGPT

In this section, we conduct a case study using ChatGPT (May 24, 2023 version) through the platform[10] provided by OpenAI. We provide multiple examples for analyzing the substantiation level of the peer review in Table 5 with different prompting techniques.

**Zero-shot prompting**.
In this case, ChatGPT is directly inquired to deal with the tasks of claim extraction and evidence linkage, without providing any prior information, results can be found in Table 6 and Table 7, respectively.

For the claim extraction task, we observe that ChatGPT cannot distinguish between subjective claims and evidence. It mistakes the facts supporting the claim as also being claims.

For the evidence linkage task, ChatGPT fails completely. It only repeats the claim again, adding along some irrelevant information. This might be connected to its inability to distinguish between claims and evidence in the first place.

**Zero-shot prompting with task descriptions**.
In this case, we additionally provide the task descriptions in our prompts, more specifically, the definition section of claim and evidence from our annotation guideline (see Table 12).

Results in Table 8 show that ChatGPT achieved a much better performance on the claim extraction task than the previous zero-shot case, all negative claims are correctly extracted.

Results in Table 9 demonstrate that despite having comprehensive task descriptions, the process of evidence linkage continues to pose significant challenges. In the output list of evidence, only the first one is relevant to the claim of interest.

**Few-shot prompting with task descriptions**.
In this case, on the basis of the previous prompts, we give ChatGPT more information by providing expected claim and evidence extractions for one example review (see Table 13).

Based on the data presented in Table 10 and Table 11, we can deduce conclusions similar to the prior case. It's evident that ChatGPT struggles to differentiate between claims and evidence and to denote their relationships.

The above examples demonstrate that ChatGPT is not able to achieve a satisfactory performance, with both specifically designed zero-shot and few-shot prompts, which highlights the need of our annotated data for instruction-tuning, and the persistent superiority of classical task-specific fine-tuned model as proposed in our work.

---

[10]https://openai.com/blog/chatgpt/

The paper proposes a method to train models for Chinese word segmentation (CWS) on datasets having multiple segmentation criteria.

-Strengths:

1. Multi-criteria learning is interesting and promising.

2. The proposed model is also interesting and achieves a large improvement from baselines.

-Weaknesses:

1. The proposed method is not sufficiently compared with other CWS models.

The baseline model (Bi-LSTM) is proposed in [1] and [2].However, these model is proposed not for CWS but for POS tagging and NE tagging. The description "In this paper, we employ the state-of-the-art architecture ..." (in Section 2) is misleading.

2. The purpose of experiments in Section 6.4 is unclear. In Sec. 6.4, the purpose is that investigating "datasets in traditional Chinese and simplified Chinese could help each other." However, in the experimental setting, the model is separately trained on simplified Chinese and traditional Chinese, and the shared parameters are fixed after training on simplified Chinese.

- General Discussion:

The paper should be more interesting if there are more detailed discussion about the datasets that adversarial multi-criteria learning does not boost the performance.

[1] Zhiheng Huang, Wei Xu, and Kai Yu. 2015. Bidirectional lstm-crf models for sequence tagging. arXiv preprint arXiv:1508.01991.

[2] Xuezhe Ma and Eduard Hovy. 2016. End-to-end sequence labeling via bi-directional lstm-cnns-crf. arXiv preprint arXiv:1603.01354.

Table 5: Example review for case study.

***Extract the negative subjective claims from the following review:***
[Review 1]

Output:

Subjective negative claims:

1. "The proposed method is not sufficiently compared with other CWS models."

2. "The baseline model (Bi-LSTM) is proposed in [1] and [2]. However, these model is proposed not for CWS but for POS tagging and NE tagging."

3. "The description 'In this paper, we employ the state-of-the-art architecture...' (in Section 2) is misleading."

4. "The purpose of experiments in Section 6.4 is unclear."

5. "In Sec. 6.4, the purpose is that investigating 'datasets in traditional Chinese and simplified Chinese could help each other.' However, in the experimental setting, the model is separately trained on simplified Chinese and traditional Chinese, and the shared parameters are fixed after training on simplified Chinese."

6. "The paper should be more interesting if there are more detailed discussion about the datasets that adversarial multi-criteria learning does not boost the performance."

Table 6: An example of ChatGPT performing the claim extraction task (zero-shot prompting).

| |
|---|
| *Extract the evidence for the claim* 
 *"The proposed method is not sufficiently compared with other CWS models."* 
 *from the following review:* 
 [Review 1] |
| Output: 
 The evidence for the claim "The proposed method is not sufficiently compared with other CWS models" in this review is the statement made by the reviewer that "The proposed method is not sufficiently compared with other CWS models." This claim is presented as a weakness of the paper, indicating that the reviewer believes that the paper's lack of comparison to other CWS models is a negative aspect of the research. |

Table 7: An example of ChatGPT performing the evidence linkage task (zero-shot prompting).

| |
|---|
| You will be given a guideline for annotating / extracting subjective claims and their relative evidence (as text spans) in scientific peer reviews, to evaluate the level of substantiation. 
 Annotation guideline: 
 [Guideline] 
 *Extract the negative subjective claims only (not evidence) from the following review:* 
 [Review 1] |
| Output: 
 Negative subjective claims: 
 1. The proposed method is not sufficiently compared with other CWS models. 
 2. The description "In this paper, we employ the state-of-the-art architecture ..." (in Section 2) is misleading. 
 3. The purpose of experiments in Section 6.4 is unclear. 
 4. The paper should be more interesting if there are more detailed discussions about the datasets that adversarial multi-criteria learning does not boost the performance. |

Table 8: An example of ChatGPT performing the claim extraction task (zero-shot prompting with task descriptions).

| |
|---|
| You will be given a guideline for annotating / extracting subjective claims and their relative evidence (as text spans) in scientific peer reviews, to evaluate the level of substantiation. 
 Annotation guideline: 
 [Guideline] 
 *Extract the evidence for the claim* 
 *"The proposed method is not sufficiently compared with other CWS models."* 
 *from the following review:* 
 [Review 1] |
| Output: 
 Claim: 1. The proposed method is not sufficiently compared with other CWS models. 
 Evidence: 
 1. The baseline model (Bi-LSTM) is proposed in [1] and [2]. However, these models are proposed not for CWS but for POS tagging and NE tagging. 
 2. The description "In this paper, we employ the state-of-the-art architecture ..." (in Section 2) is misleading. 
 3. The purpose of experiments in Section 6.4 is unclear. In Sec. 6.4, the purpose is investigating "datasets in traditional Chinese and simplified Chinese could help each other." However, in the experimental setting, the model is separately trained on simplified Chinese and traditional Chinese, and the shared parameters are fixed after training on simplified Chinese. |

Table 9: An example of ChatGPT performing the evidence linkage task (zero-shot prompting with task descriptions).

You will be given a guideline for annotating / extracting subjective claims and their relative evidence (as text spans) in scientific peer reviews, to evaluate the level of substantiation.
Annotation guideline:
[Guideline]
Annotation example for one review:
[Example Review]
For the above example review, the extracted claims and evidence are:
Positive claim 1: [Example Claim 1] Evidence 1: [Example Evidence 1]
Negative claim 2: [Example Claim 2] Evidence 2: [Example Evidence 2]
Negative claim 3: [Example Claim 3] Evidence 3: [Example Evidence 3]
***Extract the negative subjective claims only (not evidence) from the following review:***
[Review 1]

Output:
Negative claim 1: The proposed method is not sufficiently compared with other CWS models.
Negative claim 2: The purpose of experiments in Section 6.4 is unclear.
General negative claim 3: The paper should be more interesting if there are more detailed discussion about the datasets that adversarial multi-criteria learning does not boost the performance.

Table 10: An example of ChatGPT performing the claim extraction task (few-shot prompting with task descriptions).

You will be given a guideline for annotating / extracting subjective claims and their relative evidence (as text spans) in scientific peer reviews, to evaluate the level of substantiation.
Annotation guideline:
[Guideline]
Annotation example for one review:
[Example Review]
For the above example review, the extracted claims and evidence are:
Positive claim 1: [Example Claim 1] Evidence 1: [Example Evidence 1]
Negative claim 2: [Example Claim 2] Evidence 2: [Example Evidence 2]
Negative claim 3: [Example Claim 3] Evidence 3: [Example Evidence 3]
***Extract the evidence for the claim***
***"The proposed method is not sufficiently compared with other CWS models."***
***from the following review:***
[Review 1]

Output:
Claim 1: The proposed method is not sufficiently compared with other CWS models.
Evidence 1: The baseline model (Bi-LSTM) is proposed in [1] and [2]. However, these models are proposed not for CWS but for POS tagging and NE tagging.
Evidence 2: The description "In this paper, we employ the state-of-the-art architecture ..." (in Section 2) is misleading.

Table 11: An example of ChatGPT performing the evidence linkage task (few-shot prompting with task descriptions).

1. Annotate claims (Def: the reviewer's evaluation of the research, related to the paper acceptance/rejection decision-making process).
- We only annotate claims that contain subjective judgements.
- Subjectivity can be expressed explicitly by descriptions of the writer's mental state, such as in "I'm not convinced of many of the technical details in the paper" and in "I disagree a little bit here". It can also be expressed implicitly by opinionated descriptions of the work as in "There is no clear definition given for what this means" and "This paper lacks quite a bit in comparison with existing work". Both the adjective "clear" and the verb "lack" indicate subjective opinions and need to be further justified.
- Two claims should be separated if they are evaluating different aspects of the paper.
- Claims should be numbered according to their order of appearance.

2. Annotate evidence (Def: grounds/warrant/backing/qualifier that the reviewer expressed to support the above claim).
- Not all claims have a premise.
- Evidence do not have to be correct.
- Evidence can appear both before and after the evaluation.
- Evidence should be numbered according to the evaluation they support.

Table 12: Guideline for prompts.

summary_of_strengths
How to deal with negation semantic is one of the most fundamental and important issues in NLU, which is especially often ignored by existing models. This paper verifies the significance of the problem on multiple datasets, and in particular, proposes to divide the negations into important and unimportant types and analyzes them (Table 2). The work of the paper is comprehensive and solid.
summary_of_weaknesses
However, I think the innovation of this paper is general. The influence of negation expressions on NLP/NLU tasks has been widely proposed in many specialized studies, as well as in the case/error analysis of many NLP/NLU tasks. In my opinion, this paper is the only integration of these points of view and does not provide deeper insights to inspire audiences in related fields.
[Example Claim 1]
The work of the paper is comprehensive and solid.
[Example Evidence 1]
This paper verifies the significance of the problem on multiple datasets, and in particular, proposes to divide the negations into important and unimportant types and analyzes them (Table 2).
[Example Claim 2]
However, I think the innovation of this paper is general.
[Example Evidence 2]
The influence of negation expressions on NLP/NLU tasks has been widely proposed in many specialized studies, as well as in the case/error analysis of many NLP/NLU tasks.
[Example Claim 3]
does not provide deeper insights to inspire audiences in related fields
[Example Evidence 3]
this paper is the only integration of these points of view

Table 13: Example review, claims, and evidence for prompts.