# OpenReview forum: "Automatic Analysis of Substantiation in Scientific Peer Reviews"
_EMNLP/2023/Conference — EMNLP 2023 Findings_

### Official Review · Reviewer_3y4q · 2023-08-04

**Soundness:** 2

**Excitement:**

2: Mediocre: This paper makes marginal contributions (vs non-contemporaneous work), so I would rather not see it in the conference.

**Paper Topic And Main Contributions:**

The paper solves the problem of enforcing review quality by measuring the substantiveness of the claims made in reviews. To further study the problem, a new dataset is created that provides claim-evidence pairs for 550 reviews. This dataset is used to set up a baseline for this task using different models.

**Reasons To Accept:**

- The problem is very relevant in the scientific community as more and more papers are getting published.
- Dataset opens doors for new research in this often ignored area
- Paper is very well written and is a good read especially to understand how review quality can be measured

**Reasons To Reject:**

- Even though the new dataset opens up a lot of opportunities for research in this area, 550 reviews in the area of NLP are very limited, especially considering that these international conferences and reviewers from different regions have different styles. Not many details have been provided on the characteristics of reviews and reviewers. This part is also missing in the evaluation.
- Baseline achieves lower performance. It is not clear if that is due to the limited size of the dataset or the nature of the problem
- It seems intuitive to measure the substantiation by counting the number of claims with evidence, there is no systematic proof or any study that proves that this property is always true.


**Reproducibility:**

5: Could easily reproduce the results.

**Reviewer Confidence:**

3: Pretty sure, but there's a chance I missed something. Although I have a good feel for this area in general, I did not carefully check the paper's details, e.g., the math, experimental design, or novelty.

---

> ### Author Rebuttal · Authors · 2023-08-28
>
> We thank the reviewer for acknowledging that our paper is well written, that it addresses an important topic and opens up opportunities for novel research in this underdeveloped area. However, there seems to be misunderstandings concerning key elements of our paper which we would like to clarify.
>
> 1. While the size of our dataset is limited, the results in our paper demonstrate our dataset to be sufficient for fine-tuning BERT-based models. Moreover, in future work, our dataset could also be used for instruction tuning and in-context learning, where only a small amount of high-quality human curated data is required. Therefore, there are clear use-cases for our dataset of limited size. We furthermore want to add that recently published annotated datasets in this domain contain fewer samples than our SubstanReview dataset [1]. Statistics of the reviews from our annotations are shown in Table 2. For other details of the reviews, please refer to the original NLPeer paper [2], we cannot include all of them due to space limitations. Unfortunately, no information regarding the reviewers is publicly released.
>
> 2. The baseline is not fine-tuned on our dataset and thus expected to achieve lower performance than the fine-tuned models. We introduce the baseline in Lines 536 –549, it is a heuristic approach based on an open-source sentiment classifier and BERTScore. In Table 3, BERT, RoBERTa, SciBERT and SpanBERT are fine-tuned on our dataset while the baseline is not. The fact that models fine-tuned on SubstanReview significantly outperform the  baseline prove that our dataset is efficient for the claim-evidence pair extraction task. We will further emphasize and clarify this point in the final version of our paper.
>
> 3. We have conducted a systematic study to validate our intuition, please refer to Section 4.4 SubstanScore. There, we calculate the correlation between human annotated substantiation scores and SubstanScore derived from our intuition, resulting in a significant Spearman’s ρ = 0.7568.
>
> [1] Fromm, Michael, et al. "Argument mining driven analysis of peer-reviews." AAAI 2021.
>
> [2] Dycke, Nils, Ilia Kuznetsov, and Iryna Gurevych. "NLPEER: A Unified Resource for the Computational Study of Peer Review." ACL 2023.

---

### Official Review · Reviewer_7zci · 2023-08-04

**Soundness:** 4

**Excitement:**

4: Strong: This paper deepens the understanding of some phenomenon or lowers the barriers to an existing research direction.

**Paper Topic And Main Contributions:**

This paper addresses the problem of problematic reviews in NLP (not AI as suggested) conferences. It proposes an automatic quality control measure focusing on the aspect of "substantiation" (the level of support and evidence for the claims made in a paper review). The paper formulates the problem as claim-evidence pair extraction in scientific peer reviews and introduces a new, very curated dataset, which should be the first annotated dataset for this task. The dataset is not large in the "2023" sense, with 550 reviews from NLP conferences, annotated by domain experts, however, this is a good starting point for a dataset requiring such a level of specialized knowledge to be annotated.

Contributions:
- The paper introduces the SubstanReview dataset, which is a valuable resource for studying the quality of peer reviews in NLP conferences.
- The paper develops a system trained on the new dataset. It can automatically analyze the level of substantiation in reviews, enabling automatic quality control measures for problematic reviews.
- Problematic peer reviews can have serious consequences for the scientific community, leading to incorrect decisions on paper acceptance or rejection. By focusing on substantiation and providing an automated "solution" for the issue.



**Questions For The Authors:**

A. Could you provide any justification for using a sentiment classifier based on tweets for the baseline?

B. How do you plan to address the potentially limited generalizability of your approach and findings? Have you considered any strategies to adapt or validate your approach on reviews from other research domains to ensure broader applicability?

**Reasons To Accept:**

1. The paper introduces a new dataset which is the first annotated dataset for this task. This dataset has been carefully curated and annotated by domain experts, making it a valuable resource for the NLP community.
2. The paper provides a clear and detailed description of the data collection guidelines used to create the SubstanReview dataset. Additionally, the paper includes information on IAAs for both rounds of annotations, demonstrating the rigor and reliability of the dataset.
3. The task of evaluating the level of substantiation in peer reviews is highly relevant to the NLP community.
4. While the paper focuses on the NLP domain, the concept of claim-evidence pair extraction and substantiation evaluation can be applicable to other research fields beyond NLP.

**Reasons To Reject:**

1. The evaluation is done on a quite limited and domain-specific data collection. This is acceptable for the task itself but the third claim on the contribution must be better clarified (e.g., "We provide meaningful insights into the current level of substantiation in scientific peer reviews." is only valid for a specific domain and limited data points)

No other major issues were identified in the paper, just some minor remarks:
1. The sentiment classifier used in the paper is based on tweets, this could potentially bias the baseline.
2. The models used in the experiments are identified by (I suppose) HF hub tags, this should be explicitly mentioned.

**Reproducibility:**

5: Could easily reproduce the results.

**Reviewer Confidence:**

4: Quite sure. I tried to check the important points carefully. It's unlikely, though conceivable, that I missed something that should affect my ratings.

---

> ### Author Rebuttal · Authors · 2023-08-28
>
> We would like to thank the reviewer for their insightful comments. We are extremely glad that you acknowledge the careful curation of our dataset and find it to be valuable despite its limitation in scope. This exactly corresponds to our considerations in the creation of the dataset. We further reply to your suggestions in detail.
>
> * We agree that the third claim on the contribution should be refined. We will change it to “We provide meaningful insights into the current level of substantiation in publicly available peer reviews from recent NLP conferences.”
>
> * We will clarify that the models used in the experiments are identified by hugging face tags.
>
> * **Question A**:
> In fact, sentiment classification for peer review sentences is a simple task because of the explicit expressions of opinions. As demonstrated by the confusion matrix in Figure 2, positive and negative classes are very rarely confused. This task is thus unlikely to be a bottleneck in the baseline’s performance. To the best of our knowledge, there does not exist an open-source sentiment classifier trained on scientific text. Therefore, our choice to make use of a high-performing open-source sentiment classifier trained on tweets [1], appears to be suitable to us and potential bias introduced by the training on tweets should be of limited impact.
>
> * **Question B**:
> We believe that the formulation of the claim-evidence pair extraction task is generalizable to other research domains since the underlying argumentation theory is shared. However, the models would need to be further fine-tuned with domain-specific annotations since the reviewing format and terminology could be very different. For future work, we could extend our annotations to the F1000-22 subpart of the NLPeer dataset [2] which covers a wide range of research communities from scientific policy research to medicine and public health. This would require us to collaborate with domain experts from the corresponding fields.
>
> [1] Barbieri, Francesco, et al. "TweetEval: Unified Benchmark and Comparative Evaluation for Tweet Classification." Findings of EMNLP 2020.
>
> [2] Dycke, Nils, Ilia Kuznetsov, and Iryna Gurevych. "NLPEER: A Unified Resource for the Computational Study of Peer Review." ACL 2023.

---

### Official Review · Reviewer_M4ST · 2023-08-11

**Soundness:** 4

**Excitement:**

4: Strong: This paper deepens the understanding of some phenomenon or lowers the barriers to an existing research direction.

**Paper Topic And Main Contributions:**

This paper proposes a study of automatic claim substantiation evaluation in scientific reviews, and introduces a dataset of scientific reviews annotated by marking spans of text containing claims and evidence. They also propose a method for automatically pairing claims and evidence on this dataset based on an argument mining framework.

**Reasons To Accept:**

- Useful dataset (though not very large)
- Interesting and relatively novel approach based on argument mining, sound experiments

**Reasons To Reject:**

- not a major reason for rejection, but the relatively small size of the dataset is the biggest weakness

**Reproducibility:**

4: Could mostly reproduce the results, but there may be some variation because of sample variance or minor variations in their interpretation of the protocol or method.

**Reviewer Confidence:**

2: Willing to defend my evaluation, but it is fairly likely that I missed some details, didn't understand some central points, or can't be sure about the novelty of the work.

**Typos Grammar Style And Presentation Improvements:**

- line 56: "domain knowledgeability"  could be simply "domain knowledge"?
-

---

> ### Author Rebuttal · Authors · 2023-08-28
>
> We thank the reviewer for their positive assessment of our work. We are glad that you found our dataset to be useful, approach to be novel/interesting and experiments to be sound. Regarding the size of the dataset, we agree that it is not large in the sense of today’s “big data era”. However, our experiments have proven that our dataset is efficient for the claim-evidence pair extraction task. We focus on quality rather than quantity, our dataset is very carefully curated. In future work, our dataset could also be used for instruction tuning and in-context learning, where only a small amount of high-quality human curated data is required. We will make sure to add this point to our revised manuscript.

---

### Official Review · Reviewer_JsBA · 2023-08-12

**Soundness:** 4

**Excitement:**

4: Strong: This paper deepens the understanding of some phenomenon or lowers the barriers to an existing research direction.

**Paper Topic And Main Contributions:**

The paper is about automatically assigning the level of substantiation of a peer review, i.e determining whether the claims in a review are sufficiently supported by evidence. To do so, the authors formulate the task as claim-evidence pair extraction from the peer review.  The task consists of two steps: claim tagging (claim detection with binary sentiment label), and evidence linkage (for each detected claim in the previous step, retrieve the contiguous span of text from the review which supports each claim).

**Reasons To Accept:**

- An expert-annotated dataset of 550 peer reviews with paired claim and evidence spans. While useful for the specific task framed by the paper, this dataset could potentially be used by the broader argument mining research area.
- Data analysis of peer reviews from 2016 to 2022. The decrease in supported claims in peer reviews is a very interesting result (though not new/original, previous work has shown similar results, as stated by the authors).
- Argument mining system usefully adapted to the problem at hand, framed as two consecutive tasks: claim tagging (token classification task) and evidence linkage (question answering task)


**Reasons To Reject:**

- No specific reason to reject.

**Reproducibility:**

4: Could mostly reproduce the results, but there may be some variation because of sample variance or minor variations in their interpretation of the protocol or method.

**Reviewer Confidence:**

3: Pretty sure, but there's a chance I missed something. Although I have a good feel for this area in general, I did not carefully check the paper's details, e.g., the math, experimental design, or novelty.

---

> ### Author Rebuttal · Authors · 2023-08-28
>
> We thank the reviewer for their positive assessment of our work. We are glad that you found our annotated dataset to be valuable, the developed argument mining system to be useful, and the results of our analysis to be interesting.

---

### Official Review · Reviewer_td1b · 2023-08-20

**Typos Grammar Style And Presentation Improvements:** 1. Line 338-348 too long sentences, a…
**Soundness:** 4

**Excitement:**

3: Ambivalent: It has merits (e.g., it reports state-of-the-art results, the idea is nice), but there are key weaknesses (e.g., it describes incremental work), and it can significantly benefit from another round of revision. However, I won't object to accepting it if my co-reviewers champion it.

**Missing References:**

NA

**Paper Topic And Main Contributions:**

The paper studies substantiation which is one of the important aspects in the domain of peer reviews. The paper models the problem as a claim-evidence extraction task. The paper is easy to follow. A new dataset for this task is proposed and argument mining systems developed to address this task.

**Questions For The Authors:**

1. Check the reasons to rejct
2. Table 2 seems difficult to understand. What are the total no of positive/negative claims rather than % ?
3.  Table 3 for evidence linkage can you also report Precisn, Recall?
4. For table 3 and 4 since the dataset is annotated by different annotators, it is difficult to understand what is the upper bound of performance. Can you justify this?

**Reasons To Accept:**

1. The paper address one of the important issues in peer reviewing, the substantiation or specificity of reviews.
2. A new dataset and baselines are proposed for the task
3. Sufficient amount of analysis has been performed and the insights are interesting.

**Reasons To Reject:**

1. the dataset developed in the process does not seem to be very high quality. "10% of the dataset (55 reviews) was used to
calculate inter-annotator agreement (IAA)" seems too low a number to judge the quality of the dataset.
2. "Following the pilot round, our main annotation study was carried out by three expert annotators who are coauthors of this paper. They are all graduate/post-graduate NLP researchers". The annotators being the authors of the paper who created the guidelines seem to bring in a lot of bias in the dataset.
3. "Moreover, the annotators only went through a 2-hour training session, thus not qualifying as experts for this task"- The training time seems too less.
3. The substantiation score introduced in the paper is never discussed further in the paper other than Sec 4.4

**Reproducibility:**

5: Could easily reproduce the results.

**Reviewer Confidence:**

4: Quite sure. I tried to check the important points carefully. It's unlikely, though conceivable, that I missed something that should affect my ratings.

---

> ### Author Rebuttal · Authors · 2023-08-28
>
> We thank the reviewer for appreciating the motivation, writing and analysis of our paper. We would like to resolve several misunderstandings, especially regarding the annotation process. We reply to your comments below.
>
> **Response to “Reasons to reject”:**
> 1. We disagree on the point that the quality of our dataset is low as a result of 10% of reviews being used to estimate the IAA. Firstly, in the only prior work in which peer reviews were annotated at the token level [1], IAA was estimated on the basis of the entire data set containing only 77 reviews. 10% of our comparatively large dataset, containing 550 reviews, is therefore of the same order of magnitude as previously published work. We furthermore want to mention that reviews from the different conferences are proportionally represented in our 10% sample. This stratification according to the originating conference allows us to use our sample to estimate the IAA across the data set as accurately as possible.
>
> 2. There exists bias in all human annotations, it is only possible to mitigate it but not completely avoid it. While we cannot claim that there is no bias in our dataset, we do not believe that including expert annotators as co authors introduces any additional bias. In fact, it is ethical to include them as co authors since one of the key contributions of the paper is the dataset which they created. The annotation guidelines were initially established by the main author of the paper, the expert annotators participated in making adjustments through providing feedback and suggestions. Their involvement helps to identify potential areas of ambiguity and reduces the chance of misinterpretation. This is recommended practice for improving annotation quality [2].
>
> 3. This is a misunderstanding. The quoted sentence is from the description of the pilot annotation round. The annotations from this round were not included in the final SubstanReview dataset (as written in Lines 376 –377). We include this section only for completeness and to prove that a small number of expert annotators are much better than a large number of non-experts for such a complicated task. We will further emphasize this in the final version of our paper. The expert annotators who created SubstanReview have gone through approximately 20 hours of training/discussion spread over a month before proceeding to the final annotations. We will also add this to the revised manuscript.
>
> 4. Thank you for bringing this to our attention. We will make sure to also mention SubstanScore in the Introduction and Conclusion sections of our revised manuscript. Feel free to let us know if you believe further discussion of the score in other parts of the paper to be beneficial.
>
> **Response to “Questions for the authors”:**
> 1. Please refer to previous responses.
>
> 2. The total number of positive/negative claims are reported in the second and third columns of Table 2. **#Claims** stands for the *number of claims*, **%Supported claims** stands for the *percentage of claims that are paired with evidence*. **Pos** means *positive* and **Neg** means *negative*. In the final version of the paper, we will add these explanations to the caption.
>
> 3. The evidence linkage step is treated as an extractive question answering task, the common metrics to report for this task are Exact Match and F1. However, we agree that we should add the precision and recall scores for consistency. Please see the requested metrics displayed in the table below.
>
> |          	| Precision 	| Recall 	|
> |----------|-----------|--------|
> | BERT     	| 75.99     	| 80.48  	|
> | RoBERTa  	| 79.48     	| 81.03  	|
> | SciBERT  	| 79.09     	| 81.03  	|
> | SpanBERT 	| 82.39     	| 81.77  	|
> | Baseline 	| 12.19     	| 9.67   	|
>
> 4. This is another misunderstanding. Table 3 and 4 result from the same annotations (the main round). The difference between the two tables is that Table 4 takes into account error propagation while Table 3 does not. The two subtasks (claim tagging and evidence linkage) are consecutive. In Table 3, the evidence linkage step is performed on ground truth claims, whereas in Table 4, evidence linkage is performed on top of claims tagged by the system. As expected, the performance in Table 3 is better than that in Table 4.
>
> [1] Fromm, Michael, et al. "Argument mining driven analysis of peer-reviews." AAAI 2021.
>
> [2] Klie, Jan-Christoph, Richard Eckart de Castilho, and Iryna Gurevych. "Analyzing Dataset Annotation Quality Management in the Wild." arXiv preprint arXiv:2307.08153(2023).

---

### Meta-Review · Area_Chair_LnTp · 2023-09-16

**Recommendation:** 4

**Metareview:**

The paper under review addresses a key issue in the scientific community - improving the quality of peer reviews. It focuses on the aspect of substantiation, which is defined as the degree to which claims in a review are supported by evidence. The authors frame the issue as a claim-evidence extraction task and introduce a new dataset (SubstanReview) for this purpose. This dataset, although small and domain-specific (550 reviews from NLP conferences), is the first of its kind, and the reviewers acknowledge its potential value to the NLP community. The creation of this dataset and the development of a system to automatically analyze the level of substantiation in reviews are the main contributions of the paper. The reviewers generally agree that these are significant and valuable contributions. They also commend the clarity of the paper and its potential applicability beyond the task-specific domain.

However, the reviewers also raise some concerns. Firstly, the reviewer's main concern was on dataset's limitedness being applied to NLP models. Secondly, the evaluation of the system is seen as limited due to the domain-specific nature of the data collection. Questions were also raised about the generalizability of the findings and the potential bias in the baseline sentiment classifier, which is based on tweets. Thirdly, some of the claims made in the paper were seen as lacking sufficient support. For instance, the substantiation score is not discussed extensively in the paper, and the relationship between the number of claims with evidence and the level of substantiation is assumed without systematic proof.

Overall, while the paper makes a significant contribution to the field and could potentially impact the way peer reviews are conducted, there are several important issues that need to be addressed to enhance the robustness and generalizability of the findings. The reviewers' scores reflect these mixed feelings, with some expressing strong support and others being more ambivalent or even critical. Regardless of the concerns discussed by the reviewers, this paper has substantial merits that can affect the broader communities of NLP.

---

### Decision · Program_Chairs · 2023-10-07

**Decision:**

Accept-Findings

**Comment:**

The paper under review addresses a key issue in the scientific community - improving the quality of peer reviews. It focuses on the aspect of substantiation, which is defined as the degree to which claims in a review are supported by evidence. The authors frame the issue as a claim-evidence extraction task and introduce a new dataset (SubstanReview) for this purpose. This dataset, although small and domain-specific (550 reviews from NLP conferences), is the first of its kind, and the reviewers acknowledge its potential value to the NLP community. The creation of this dataset and the development of a system to automatically analyze the level of substantiation in reviews are the main contributions of the paper. The reviewers generally agree that these are significant and valuable contributions. They also commend the clarity of the paper and its potential applicability beyond the task-specific domain.

However, the reviewers also raise some concerns. Firstly, the reviewer's main concern was on dataset's limitedness being applied to NLP models. Secondly, the evaluation of the system is seen as limited due to the domain-specific nature of the data collection. Questions were also raised about the generalizability of the findings and the potential bias in the baseline sentiment classifier, which is based on tweets. Thirdly, some of the claims made in the paper were seen as lacking sufficient support. For instance, the substantiation score is not discussed extensively in the paper, and the relationship between the number of claims with evidence and the level of substantiation is assumed without systematic proof.

Overall, while the paper makes a significant contribution to the field and could potentially impact the way peer reviews are conducted, there are several important issues that need to be addressed to enhance the robustness and generalizability of the findings. The reviewers' scores reflect these mixed feelings, with some expressing strong support and others being more ambivalent or even critical. Regardless of the concerns discussed by the reviewers, this paper has substantial merits that can affect the broader communities of NLP.